# ILPO-MP: Mode Priors Prevent Mode Collapse when Imitating Latent Policies from Observations

**Oliver Struckmeier**                                    *oliver.struckmeier@aalto.fi*
*Intelligent Robotics Group*
*Aalto University, Espoo, Finland*

**Ville Kyrki**                                           *ville.kyrki@aalto.fi*
*Intelligent Robotics Group*
*Aalto University, Espoo, Finland*

**Reviewed on OpenReview:** *https://openreview.net/forum?id=f3JLnnZsAm*

## Abstract

Imitation learning from observations (IfO) constrains the classic imitation learning setting to cases where expert observations are easy to obtain, but no expert actions are available. Most existing IfO methods require access to task-specific cost functions or many interactions with the target environment. Learning a forward dynamics model in combination with a latent policy has been shown to solve these issues. However, the limited supervision in the IfO scenario can lead to mode collapse when learning the generative forward dynamics model and the corresponding latent policy. In this paper, we analyze the mode collapse problem in this setting and show that it is caused by a combination of deterministic expert data and bad initialization of the models. Under the assumption of piecewise continuous system dynamics, we propose ILPO-MP, a method to prevent the mode collapse using clustering of expert transitions to impose a mode prior on the generative model and the latent policy. We show that ILPO-MP prevents mode collapse and improves performance in a variety of environments.

## 1 Introduction

Imitation learning (IL) as a paradigm to imitate the policy of another agent or expert has been introduced in the 90s (Michie et al., 1990; Pomerleau, 1991; Bain & Sammut, 1995). The recent advances in machine learning and the adoption of deep learning have enabled researchers to process high-dimensional data and model more complex policies. As a result, deep learning based IL methods have been applied in various domains, such as controlling autonomous vehicles (Pomerleau, 1991; Codevilla et al., 2018; Bojarski et al., 2016; Pan et al., 2017; Giusti et al., 2015; Abbeel & Ng, 2004), robotic manipulation (Fang et al., 2019; Finn et al., 2016; Radosavovic et al., 2021) and learning to play video games and complete tasks in simulated environments (Ho & Ermon, 2016; Aytar et al., 2018). However, most state-of-the-art IL methods require a large amount of expert state-action data or rely on reinforcement learning methods which is often not possible in real-world applications.

Imitation learning from observations only (IfO) tries to circumvent the need for expert actions. To this end, Edwards et al. (2019) proposed ILPO, a semi-supervised model-based IfO method in which a latent policy is learned concurrently with a forward dynamics model conditioned on latent actions and expert state observations. Using the latent policy, a data-efficient mapping from latent actions to real action space can be learned from few environment interactions without the need for task-specific knowledge or expert actions. However, the lack of supervision through expert actions can make it difficult to recover from mode collapse, a state in which the latent dynamics model collapses and expresses the observed dynamics with only one latent action. Since forward dynamics and latent policy are learned together, no useful latent policy can

be obtained in this collapsed state. This problem makes the method sensitive to the initialization of the model parameters and becomes especially an issue when expert demonstrations have been collected using a deterministic expert policy.

This work, analyzes the previously unaddressed mode collapse problem in model-based IfO and discusses possible solutions. Furthermore, we propose ILPO-MP, an unsupervised clustering approach to pre-train the latent policy and dynamics model and alleviate mode collapse during latent policy learning. We evaluate the performance of the learned latent policies and show that pre-training can prevent mode collapse and improve the performance of the baseline ILPO approach.

## 2 Related Work

### 2.1 Imitation Learning

Imitation Learning (IL) describes methods in which an imitation policy is learned to mimic the behavior of an expert or a target agent. IL methods usually assume that expert demonstrations consist of state and action pairs. Behavioral cloning (BC) (Michie et al., 1990; Pomerleau, 1991; Bain & Sammut, 1995; Bagnell et al., 2006; Ross et al., 2011; Daftry et al., 2016), directly clones the expert policy from the expert state-action pairs. Drawbacks of BC are mainly related to covariate shift. Feedback loops and accumulating uncertainty in the imitated behavior may lead the imitator to unknown states in which the learned behavior might fail (Ross & Bagnell, 2010; Spencer et al., 2021).

The second approach to IL is based on inverse reinforcement learning (IRL), where expert demonstrations are used to infer the expert's reward function and then use reinforcement learning methods and access to the environment to learn an imitation policy. While those methods are considered more robust to covariate shift, most of the IRL-based methods require an extensive amount of environment interactions (Ng et al., 2000; Abbeel & Ng, 2004; Russell, 1998; Finn et al., 2016; Ho & Ermon, 2016).

Another approach to the imitation learning problem is aligning state and action distributions via some similarity metric such as temporal similarity of state action pairs (Schroecker & Isbell, 2017) or the Wasserstein distance (Dadashi et al., 2020; Fickinger et al., 2021). These methods, however make assumptions that source and target state-action distributions are alignable via their respective similarity metrics.

### 2.2 Imitation Learning From Observations Only

Imitation Learning from Observations Only (IfO) constrains the IL setting further by not relying on expert action information (Torabi et al., 2019) and is considered strictly harder than IL with expert actions Kidambi et al. (2021). The motivation is that precise recording of expert actions is costly or even impossible in many real-world environments. However, often there is abundant video material or other expert state recordings to learn from.

In their review, Torabi et al. (2019) categorize IfO methods into model-free and model-based approaches. Model-free approaches usually address the lack of expert actions using adversarial methods or reward engineering as substitute supervision signals. Adversarial methods (Torabi et al., 2019; Nair et al., 2017; Ho & Ermon, 2016; Schroecker & Isbell, 2017) usually require large amounts of interactions with the environment, which is not always feasible or safe. Reward engineering can address this, but requires either human knowledge of the task or other data-driven methods to extract supervision signals from the expert data (Sermanet et al., 2018; Liu et al., 2018; Gupta et al., 2017). A mixture of both approaches was presented by Aytar et al. (2018), where representations learned from videos were used to generate reward trajectories to guide the exploration of a standard Reinforcement Learning agent trained in the target environment.

Unlike model-free methods, model-based IfO methods can be used to learn efficient control policies allowing more flexible planning. Behavioral Cloning from Observation (BCO) (Torabi et al., 2018) learns an inverse dynamics model $P(a|s_{t+1}, s_t)$ to infer actions from transitions while interacting with the environment and then uses the actions to learn the imitation policy. In a very similar fashion Radosavovic et al. (2021) learn an inverse dynamics model and a policy to support the training of a Reinforcement Learning agent with

state-only expert data. While both methods show promising results, they require many interactions with the target environment to learn a dynamics model. Furthermore, Sun et al. (2019) argue that there is no guarantee that obtaining an inverse dynamics model is always possible. Consider that the inverse dynamics model can be rewritten as

$$P(a|s_{t+1}, s_t) = \frac{P(a, s_{t+1}, s_t)}{P(s_{t+1}, s_t)} = \frac{P(s_{t+1}|a, s_t)\pi^\star(a|s_t)}{P(s_{t+1}|s_t)} \tag{1}$$

Thus,

$$P(a|s_{t+1}, s_t) \propto P(s_{t+1}|a, s_t)\pi^\star(a|s_t) \tag{2}$$

shows that the inverse model is ill-defined without considering the corresponding policy, in the case of stochastic dynamics. To learn a probabilistic inverse model for a particular expert, the data to learn the model needs to be gathered from the same expert. Yang et al. (2019) term a similar issue inverse dynamics disagreement and discuss it by comparing LfO and learning from demonstration (LFD) which assumes access to expert actions. Minimising this disagreement is difficult because it requires knowledge of environment dynamics and expert actions. However, an upper bound can be derived and minimized. The authors compare their method, inverse dynamics disagreement minimization, to model-free LfO methods and show significant improvement. The theoretical and experimental results suggest that inverse dynamics models in LfO are impacted negatively by the reliance on the expert policy.

To circumvent the need for the true expert policy, Edwards et al. (2019) have proposed ILPO using a forward dynamics model, to the best of our knowledge the first and only method taking this approach. A latent policy $\pi_\omega(z|s_t)$ is learned in parallel with a latent forward dynamics model $G_\theta(s_{t+1}|s_t, z)$ where $\omega$ and $\theta$ parameterise the models. Later the latent policy is mapped to the real action space with few environment interactions. Those two steps are referred to as (1) *latent policy learning* and (2) *action remapping*. The dependency on the expert policy is approximated by

$$\pi^\star(a|s_t) \sim P(a|z)\pi_\omega(z|s_t) \tag{3}$$

and the expert forward dynamics model by

$$P^\star(s_{t+1}|s_t, a) \sim G_\theta(s_{t+1}|s_t, z)\pi_\omega(z|s). \tag{4}$$

Note that ILPO assumes a discrete action space and therefore a discrete latent action space. In summary, the reliance on the expert policy is split into an intermediate learning problem in which a low dimensional latent action space $Z$ is learned in an unsupervised way from offline expert state observations.

In practice, the forward dynamics model $G_\theta$ is trained by predicting $|Z|$ possible state transitions $\Delta_z = s_{t+1} - s_t$ and picking the best prediction. The minimization loss

$$\mathcal{L}_{\min} = \min_z ||\Delta_t - G_\theta(s_t, z)||^2 \tag{5}$$

assures gradient updates are performed on the instance of $G$ which leads to the best prediction to learn to separate the different modes/latent actions. The latent policy is trained by using the forward model and minimizing the loss

$$\mathcal{L}_{\exp} = ||\Delta_t - \sum_z \pi_\omega(z|s_t)G_\theta(s_t, z)||^2 \tag{6}$$

training the latent policy to predict probabilities that explain the observed transitions.

Ideally $Z$ should be low dimensional such that the behavior of the expert can be sufficiently expressed with the available latent actions while the mapping to real actions $P(a|z)$ is as easy as possible to learn. Edwards et al. (2019) have investigated by hyperparameter search that a good initial guess for the number of latent actions is the true number of actions that the expert takes.

## 3 Mode collapse when imitating latent policies

ILPO promises high data efficiency compared to model-free and even many model-based LfO methods with respect to the number environment interactions. However, it can not always guarantee that a good latent

policy can be learned. A core component of ILPO, the forward dynamics model, which is a generative model, is susceptible to mode collapse, which has not been previously discussed.

### 3.1 Mode Collapse in Generative Models

Mode collapse, usually discussed in the context of Generative Adversarial Neural Networks (GANs), describes a failure scenario in which a multimodal generative model collapses to one mode, and the generator network generates data with low variety (Salimans et al., 2016; Che et al., 2016). Investigations of the mode collapse problem have shown that it is related to catastrophic forgetting and the optimization process in GANs, which prevents the generator from breaking out of the model collapse (Che et al., 2016; Thanh-Tung & Tran, 2020). A symptom of mode collapse is that the generator fails to generate diverse data, which is well separated in observation space from the training data. As a consequence this also prevents the discriminator from learning meaningful features.

Solutions to mode collapse in GANs include clustering the data based on knowledge about the classes in the training dataset (mode priors), mode regularization, minibatch discrimination (Che et al., 2016), continual learning or using optimizers with momentum to propagate knowledge during training to prevent catastrophic forgetting (Thanh-Tung & Tran, 2020).

### 3.2 Observing mode collapse

In ILPO, mode collapse occurs when the learned forward dynamics model can express the expert observations with few latent actions or, worst case, collapses to one mode. A unimodal latent action cannot be remapped to the real action space successfully. A deterministic expert policy $a = \pi^\star(s_t)$ is especially susceptible to mode collapse. In this case, the approximation of a deterministic expert policy can be simplified to

$$a = \pi^\star(s_t) \sim P(a|z)\pi(z|s_t) = P(a|\pi(s_t)) \tag{7}$$

which is independent of z. Furthermore, the forward dynamics can be simplified to

$$\begin{aligned} G(s_{t+1}|s_t, z) = & \frac{G(s_{t+1}, s_t, z)}{G(z, s_t)} = \\ & \frac{G(z)G(s_{t+1}, s_t)}{G(z)G(s_t)} = G(s_{t+1}|s_t) \end{aligned} \tag{8}$$

which does not depend on the latent action. As a result, the minimization loss term in Eq. 5 collapses the mode for which by means of random initialization $L_{\min}$ is smallest. We discuss the effects of random initialisation in 6.4.

Mode collapse in ILPO can be observed by monitoring the diversity of predicted latent actions by the latent policy $\pi_\omega$ during latent policy learning. For example, let us look at the MountainCar Gym environment (Brockman et al., 2016). In Fig. 1a, we can see that the average task performance of ILPO with deterministic expert demonstrations shown in blue is equal to a random policy. The task can not be learned. Let us now demonstrate how to identify mode collapse in the latent policy learning step of ILPO. Fig. 1b shows the number of unique latent actions in a batch predicted by the latent policy at the beginning of the latent policy learning process. With deterministic expert demonstrations, the number of unique actions drops to 1 early in the training and can not recover. The forward dynamics model collapses to one mode and predicts only one latent action, which is insufficient to learn a meaningful mapping to the real action space.

Next, we demonstrate which properties of the expert demonstrations facilitate mode collapse in the latent policy learning step. Fig. 2a shows the 2D deterministic expert demonstrations in the MountainCar experiment colored by the ground truth expert actions. The state-action space exhibits distinct clusters of expert actions. Within each cluster, transitions are caused by the same expert action, and even the boundaries are clearly defined. As a result, there is very little variation in the action selection at any point in the state space. We highlight a few trajectories of the plotted expert data in the figure. These trajectories are deterministic paths through the state-action space, highlighting that the expert is never uncertain about its action selection. As a result, the dynamics in the expert data can be explained by a model that is not conditioned on actions, as shown in Eq. 8.

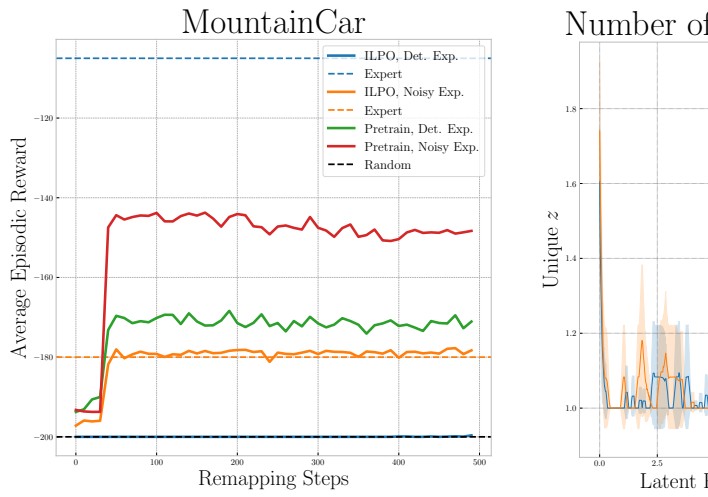
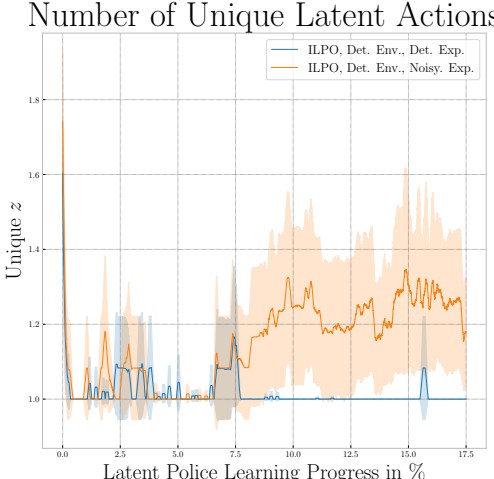

(a) MountainCar task performance when performing ILPO from deterministic and noisy expert demonstrations.

(b) Diversity of latent actions during the first 17% of the latent policy learning.

Figure 1: A demonstration of mode collapse in the MountainCar experiment and the resulting imitation performance.

# 4 Addressing Mode Collapse

In the following, we will discuss how mode collapse in imitation learning has been addressed so far and propose a new method for pre-training latent policies to prevent mode collapse.

## 4.1 Noisy Expert Demonstrations in ILPO

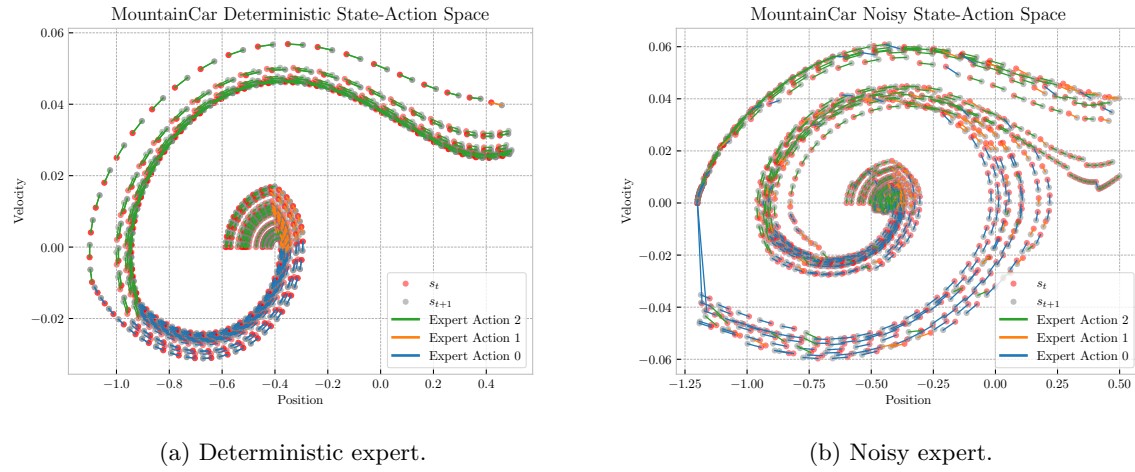

(a) Deterministic expert.

(b) Noisy expert.

Figure 2: Visualisation of the expert states, actions and transitions in the MountainCar experiment.

While mode collapse has not been discussed by Edwards et al. (2019), the authors briefly note that ILPO performs better when the expert actions are noisy and that this is usually the case when learning from humans. Fig. 2b shows a state-action space with noisy expert actions. Compared to Fig. 2a, such noisy expert demonstrations display locally diverse expert actions. The performance and latent action diversity with noisy expert actions is shown as orange in Figures 1a and 1b, the task can now be learned. This, limits

the range of applications since the noise must be part of the expert's behavior while the demonstrations are recorded. It is impossible to introduce noise to deterministic expert states after the data collection since the states are a direct result of the action selection of the expert. One major issue is that tasks like MountainCar require accurate control to solve the task, which is not always possible when the expert is uncertain in its action selection. In this example, the agent must build enough momentum over an extended period of time which is significantly more difficult and often impossible when random actions counteract the momentum. In Fig. 1a, the blue and orange dashed lines show the expert performance of the deterministic expert and the same expert with a 20%chance of taking random actions while recording the MountainCar expert[1]. The expert performance with this amount of action noise is close to random performance ($-200$ for a random policy, $-186$ with noisy actions compared to $-105$ with deterministic actions). Finally, we want to discuss the assumption in Edwards et al. (2019), that humans provide noisy demonstrations, which we think does not generally hold. Previous work has shown that humans provide noisy demonstrations in robot control tasks such as teaching robot manipulation tasks Mandlekar et al. (2021). However, demonstrations in this setting are usually continuous and concerned with low level control which is different from ILPO Edwards et al. (2019). In ILPO, noise in the action selection means that at each decision there is a probability to sample a discrete action from a uniform distribution. However, in many problems, humans select discrete actions based on the expected utility of an action following the Boltzmann distribution (see softmax action selection). For example, when uncertain, humans will usually choose the best perceived option, given the uncertain situation, not select a random action. Furthermore, humans do not select a random action when they are quite certain about the action selection. More formally, in a state $s_t$, the probability of taking action a is proportional to the value of the action and current state: $P(a|s) \propto e^{\frac{-Q(s,a)}{\lambda}}$. The temperature parameter $\lambda$ describes the uncertainty in the action selection resulting from external factors such as the skill level of the demonstrator. The Q-function usually implies that action selection becomes more uncertain at decision boundaries, especially for novices. The noise exhibited by a human expert focuses around these decision boundaries where uncertainty is high and therefore not affect the mode collapse problem. For example, let us look at Figure 9 in the manuscript. In a stochastic environment, the decision boundaries are less clear compared to Figure 2a, but the action selection is still mostly uniform across the state space, leading to mode collapse. Figure 2b shows that ILPO assumes uniform noise across the state-action space, which is a different type of noise in action selection.

In conclusion, noisy expert demonstrations alleviate the mode collapse problem but impose constraints on the expert data collection procedure. Furthermore, collecting noisy expert data using the procedure in Edwards et al. (2019) can significantly limit the capability of the expert and is not always representative of how humans make decisions.

---

**Algorithm 1** Pre-training

---

**Require:** Expert state-only data $D^e$, latent policy parameters $\omega$, latent dynamics parameters $\theta$
  1: $T^e \leftarrow GetTransitions(D^e)$
  2: $z^l \leftarrow AgglomerativeClustering(T^e)$
  3: $\omega \leftarrow E2E\ Pre\text{-}train\ \pi_\omega(D^e, z^l)$
  4: $\theta \leftarrow E2E\ Pre\text{-}train\ G_\theta(D^e, z^l)$
  5: ILPO$(D^e, \omega, \theta)$

---

### 4.2 ILPO-MP: Pre-training ILPO with a Mode Prior

To circumvent the need for noisy expert actions in ILPO, we propose ILPO-MP, a pre-training method to prime the latent policy and latent dynamics model with a mode prior such that bad initializations that lead to mode collapse are less likely. We chose mode priors over other methods of preventing mode collapse discussed in 3.1 because of their simplicity and effectiveness. Mode priors allow us to incorporate prior knowledge about the underlying modes of a expert data distribution, enabling the generative dynamics model to be guided towards producing a diverse set of latent actions.

---

[1]This is how Edwards et al. (2019) introduced noise to the expert demonstrations

But how can one obtain a prior from observations alone? The underlying dynamics of many systems, from simple physical ones to robots, can be described using differential equations like $\dot{s} = f(s, a)$, often subject to some constraint $c(s) \leq 0$. In many robotics and RL scenarios, it is assumed that $f$ and $c$ are continuous functions, meaning that small variations in s and a cause small variations in the outputs $\dot{s}$. Piecewise continuity in $f$ and $c$ relaxes this assumption to accommodate abrupt changes like, for example, contact forces in robotics. We leverage this assumption of piecewise continuous dynamics to cluster the expert demonstrations into modes where behavior remains relatively consistent. In practice, the raw expert state demonstrations usually do not permit a meaningful separation of modes. Instead, we base our mode prior on the observation that the same actions in the expert data often lead to similar *transitions* in state-space.

Table 1: Analysis of state-action and transitions-action similarity using the Procrustes Disparity between state/transition and action distance matrices. Using 5000 states from the expert data.

| Experiment | Procrustes state-action disparity | Procrustes transition-action disparity |
|---|---|---|
| Acrobot | 0.99 | 0.94 |
| Cartpole | 0.93 | 0.003 |
| MountainCar | 0.66 | 0.61 |
| LunarLander | 0.90 | 0.42 |
| Pong (vector) | 0.99 | 0.64 |

We can quantify this principle by measuring the correlation between states and true expert actions as well as transitions and true expert actions. If the assumption of piecewise continuity holds, we expect to see high correlation between transitions and actions. We obtain three pairwise distance matrices: one for expert states, one for the transitions, and one for the ground truth actions. An example of those matrices from left to right is shown in fig. 3. State and transition distance matrices are computed using the 2-norm. The action-distance matrix represents the equality between elements (actions) $i$ and $j$ and is 1 if the action an $i$ is equal to the action at $j$, and 0 otherwise. We then compute the Procrustes disparity Gower (1975);

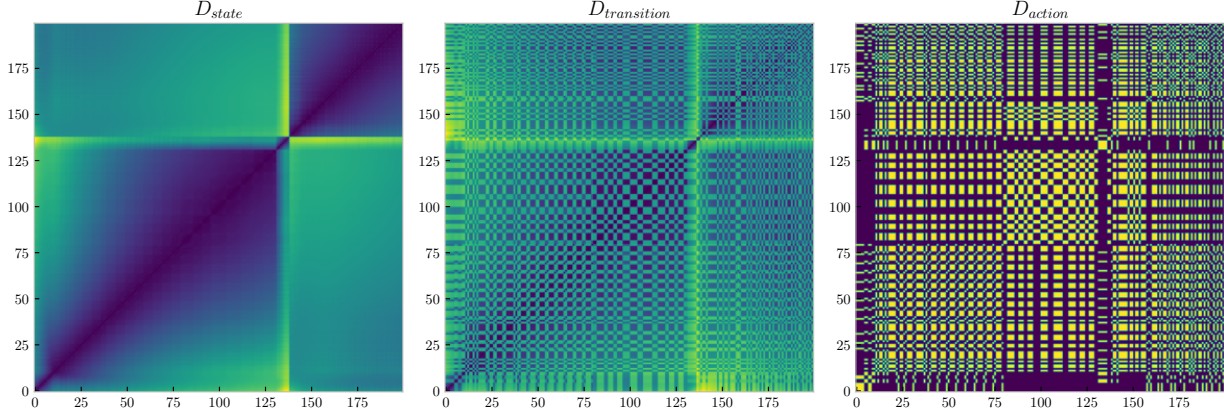

Figure 3: Visualisation of the pairwise distance matrices in the LunarLander expert data for the first 200 samples. From left to right: (left) distance matrix of states, (middle) distance matrix of transitions and (right) distance matrix of the expert actions. Qualitative analysis indicates correlation between transitions and actions.

Krzanowski (2000) for state-action and transition-action distance matrices.[2]

The results are shown in table 1. We can see that generally, the transition-action matrices have significantly lower Procrustes disparities, indicating higher similarity. The similarity assumption holds in all environments except the Acrobot environment, which we investigate in more detail in 6.2.

---

[2]Procrustes analysis aligns two sets of data points by minimising differences in translation, rotation, and scaling, by minimising the Procrustes disparity or distance which is usually a sum of the squares of the pointwise differences between two datasets.

Algorithm 1 shows how the pre-training step in ILPO-MP precedes the baseline ILPO training. We use agglomerative clustering on the expert state transitions to obtain latent action labels $z^l$ which serve as a pseudo-labels for each transition in the expert dataset.[3] The number of clusters is thereby the number of latent actions for the baseline ILPO method. Fig. 4 shows an example from the Pong environment and

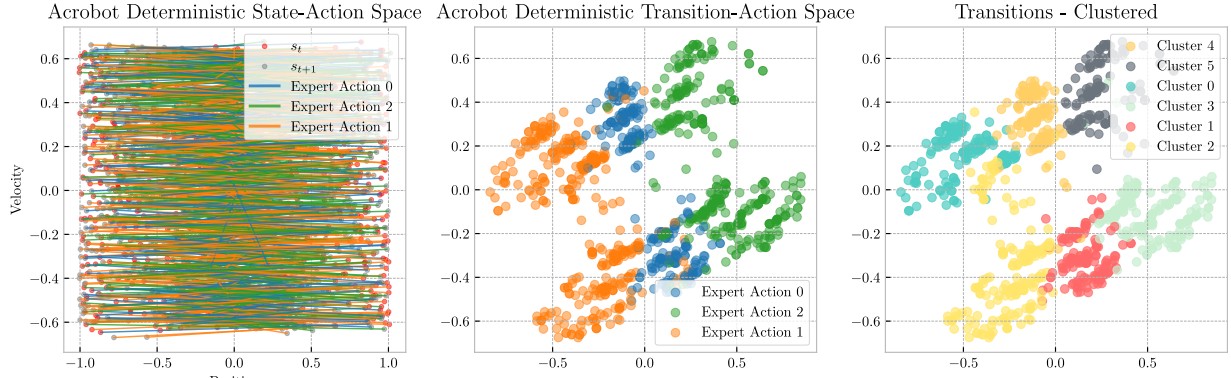

Figure 4: From left to right: (1) visualisation of transitions in state-space as lines connecting states, colored by expert action. We observe no patterns in the expert data. (2) Transitions in transition space ($\Delta_t = s_{t+1} - s_t$), colored by expert action. The higher dimensional transition data have been reduced to 2 dimensions using PCA. We can see that there are clusters where similar transitions have similar expert actions. (3) Clustering result with 6 latent actions. The clustering gives a good initialisation for the real expert actions.

highlights how in transition space clustering succeeds. The state-only expert data and the resulting labels are then used to pre-train the latent policy $\pi_\omega$ and the forward dynamics model $G_\theta$ end-to-end:

$$\mathcal{L}_{\text{pretrain}} = ||\Delta_t - G_\theta(s_t, z^l)||^2 + ||z^l - \pi_\omega(z|s_t)||^2 \tag{9}$$

ILPO-MP uses the cluster labels $z^l$ as targets to train $\pi_\omega$ and as prior to replace the minimum in Eq. 5 selecting through which mode of $G_\theta$ to backpropagate.

After performing the pre-training, the latent policy learning and remapping is performed like in the baseline ILPO method with the pre-training network weights $\omega$ and $\theta$.

## 4.3 Guided ILPO-MP

We observed that the latent policy learning can end up in a collapsed state despite the pre-training when the latent policy learning is running for many epochs. This can be observed with the tools introduced in 3.2. During a large number of latent policy learning epochs, the number of different latent actions can collapse. Longer training means more optimisation steps that modify the model parameters of $G_\theta$ and $\pi_\omega$ and as a result more chances to end up in bad local minimum in which the latent policy is collapsed. This is an issue when more training epochs are required to learn more complex feature extractors. To this end, we propose a variant of ILPO-MP to guide the latent policy learning using the prior obtained from clustering.

The proposed guided ILPO-MP approach extends the ILPO latent policy learning loss terms $\mathcal{L}_{min}$ (Eq. 5) and $\mathcal{L}_{exp}$ (Eq. 6) by the pre-training loss term presented in Eq. 9,

$$\mathcal{L}_{\text{guidance}}(e) = (1 - \frac{e}{N})\mathcal{L}_{\text{pre}} + \mathcal{L}_{\text{exp}} + \mathcal{L}_{\text{min}} \tag{10}$$

where $N$ is the total number of epochs and e is the current epoch. In practice, guided ILPO-MP trains the pre-training objective in parallel with the latent policy to prevent later collapse with the strongest regularisation applying at the beginning of the training to prevent mode collapse early in the training.

---

[3]Agglomerative clustering is a hierarchical clustering method Johnson (1967) and merges individual data points or clusters iteratively based on a measure of similarity, forming a hierarchical tree (dendrogram).

# 5 Experiments

In this section we provide more evidence for our claims that mode collapse is indeed a problem for ILPO and show how ILPO-MP prevents it. In 4 out of 6 environments, we notice that ILPO is unreliable and worst case, fails entirely. Implementation details are discussed in the Appendix in A.1.

## 5.1 Pre-training can recover mode collapse

First, let us briefly discuss the **CartPole** environment. ILPO is able to fully solve the task every single time with both deterministic and noisy expert demonstrations. No mode collapse occurs in this environment. ILPO-MP is able to reproduce the same behavior.

Recall Fig. 1, which we used this environment as an example of mode collapse in the **MountainCar** environment. Comparing ILPO (blue) and ILPO-MP (green) we can see that pre-training helped with the mode collapse where ILPO failed to learn. Although ILPO-MP can not achieve expert performance, we perform better than baseline even with noisy expert demonstrations (orange). Interestingly we find that ILPO-MP with noisy expert demonstrations (red) performs best and vastly outperforms the expert it is trying to learn from (dashed orange). We hypothesize that pre-training helped ILPO to separate the modes while ILPO alleviates the effects of the noise in the expert data. Priming the latent dynamics model with the assumption of piecewise continuous dynamics might have made the latent policy learning more robust to occasional deviations caused by random expert actions.

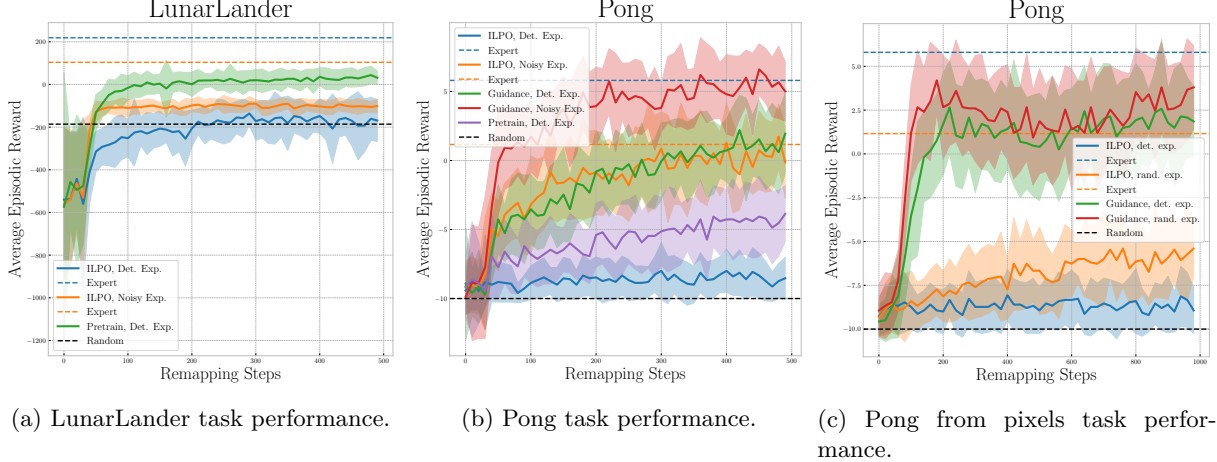

(a) LunarLander task performance.   (b) Pong task performance.   (c) Pong from pixels task performance.

Figure 5: Task performance

The results in the **LunarLander** environment presented in Fig. 5a are clear. Baseline ILPO (blue) does not outperform a random policy. Noisy expert data slightly improves task performance, but qualitative analysis does not indicate any behavior resembling the expert. ILPO-MP almost achieves the performance of the noisy expert when learning from deterministic expert actions.

The last vector-based environment is our custom built two player **Pong** environment, shown in Fig. 5b. Baseline ILPO (blue) performs poorly with deterministic expert demonstrations. Qualitative analysis of the imitated behavior shows that the agent moves randomly and sometimes gets a lucky hit and scores, resulting in performance close to a random policy. ILPO with noisy expert demonstrations (orange) achieves its respective expert performance. ILPO-MP from deterministic and noisy expert achieved better and comparable performance. In this experiment, we also found that using the guidance mode proposed in 4.3 yields further improvements. IPLO-MP with deterministic demonstrations is another significant improvement over baseline ILPO and ILPO-MP. Furthermore, ILPO-MP with noisy demonstrations achieves the best task performance and equals the deterministic expert.

The Pong from pixels experiment is based on the same expert dataset as the vector pong experiment, but now observations are in pixel space. The results are shown in Fig. 5c. Although one would expect similar

results to the vector Pong experiment, we observed that mode collapse still occurs and that ILPO with noisy expert demonstrations (orange) performs much worse. ILPO-MP performs on par with baseline ILPO. The pre-training is likely "forgotten" during the longer latent policy learning step process when compared to the simple vector-based environments. We found that instead of pre-training, the guidance approach yields better results. With our proposed guidance method, we can achieve great results in both cases, deterministic (green) and noisy (red) expert demonstrations.

In conclusion, our proposed pre-training approach leads to improved task performance and works in environments where ILPO previously failed. Furthermore, we show that learning latent policies is applicable to adversarial environments, meaning environments in which the imitator has to play against another agent that is adapting to its behavior.

## 6 Discussion

### 6.1 Complex datasets

If the expert observations $s$ are generated by an underlying physical system with generative factors $n$ like for example in vision-based tasks, the dynamics in observation space are likely not continuous. However, if the underlying physical system adheres to the piecewise continuity assumption, one can learn a mapping $n = g(s)$, which maps pixel observations into a space that approximates the true underlying dynamics $\dot{n} = f(g(s), a)$. Extending ILPO-MP to more complex environments is a task of learning good representations that capture the underlying system dynamics. Recent works such as Zhang et al. (2020) and Lu et al. (2022) have shown that invariant representations can capture task-relevant features. Other works extract meaningful representations from sequential data that capture the sparse nature of transitions in video data Klindt et al. (2020) and take into account temporal coherence in sequential data to better model latent dynamics Struckmeier et al. (2023). We illustrate this in the Pong environment, where ILPO-MP works from both, vector and pixel states (same underlying system dynamics) but with a more expressive CNN-based architecture for the pixel environment. In the future we aim to explore these representation learning methods to extend ILPO-MP to learning in more complex pixel based environments.

### 6.2 Limitations

ILPO can solve the **Acrobot** environment with ease as demonstrated by (Edwards et al., 2019), shown in blue in Fig. 6a. Our proposed method, ILPO-MP (green), achieves significantly lower task performance, which is surprising given the good performance of ILPO and the simplicity of the task. We find that the Acrobot expert exploits an implementation detail of the physics in the Acrobot environment that violates the assumption we make on the expert demonstrations discussed in 4.2. The expert has learned to spin the outer joint extremely fast so that during one step of the environment physics, multiple rotations can occur. Stiff systems, modeled by ordinary differential equations, are prone to this issue. Such systems occur when the desired solution changes slowly while nearby solutions undergo rapid variations. This effect is closely tied to the assumption of piecewise continuity we make in our pre-training method. Piecewise continuity assumes overall system continuity but acknowledges the possibility of rapid changes or discontinuities at specific points or intervals. Stiff systems often violate this assumption due to significant variations in the rates of change among variables or components.

In the case of the Acrobot environment, the states consisting of the sine and cosine of the joint angles can not express multiple rotations and discontinuities are introduced into the expert data. Fig. 6b shows the states and next states colored in red and gray and their transition colored by the ground truth action. Lines representing transitions cross the circular state space. There is no discernible continuous trajectory like in the MountainCar data (see Fig. 2a). The correlation metric for the Acrobot environment in Table 1 further confirms this.

While the issue of modeling physics in the Acrobot environment breaks the assumption we make for our pre-training approach, we argue that this does not necessarily reflect a limitation of our proposed method in general. The Acrobot environment is a simplified simulation that focuses on specific dynamics and constraints

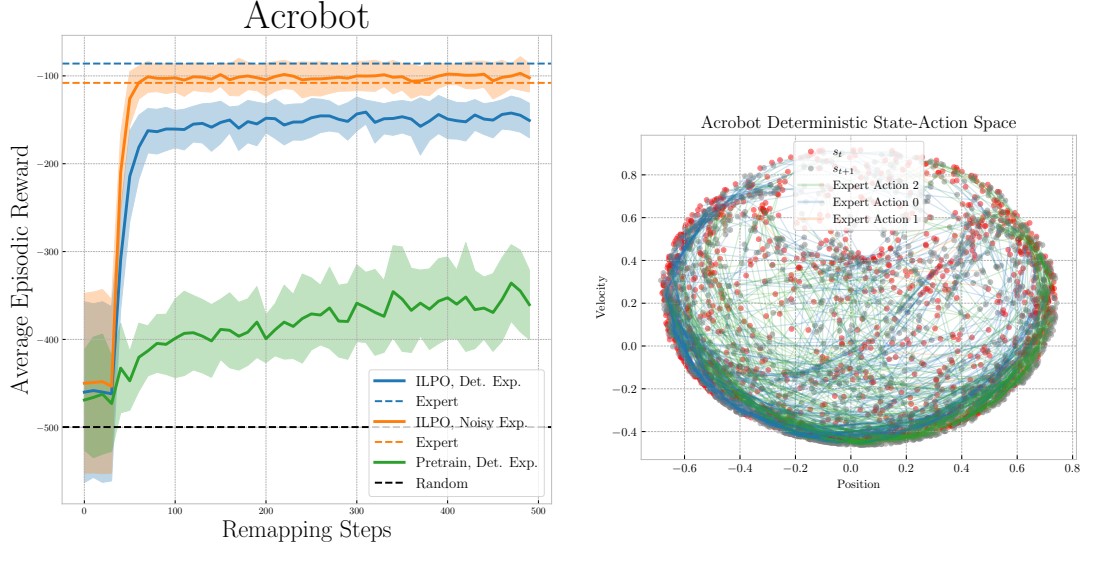

(a) Acrobot task performance.

(b) Acrobot deterministic expert data (2000 samples).

Figure 6: Acrobot results

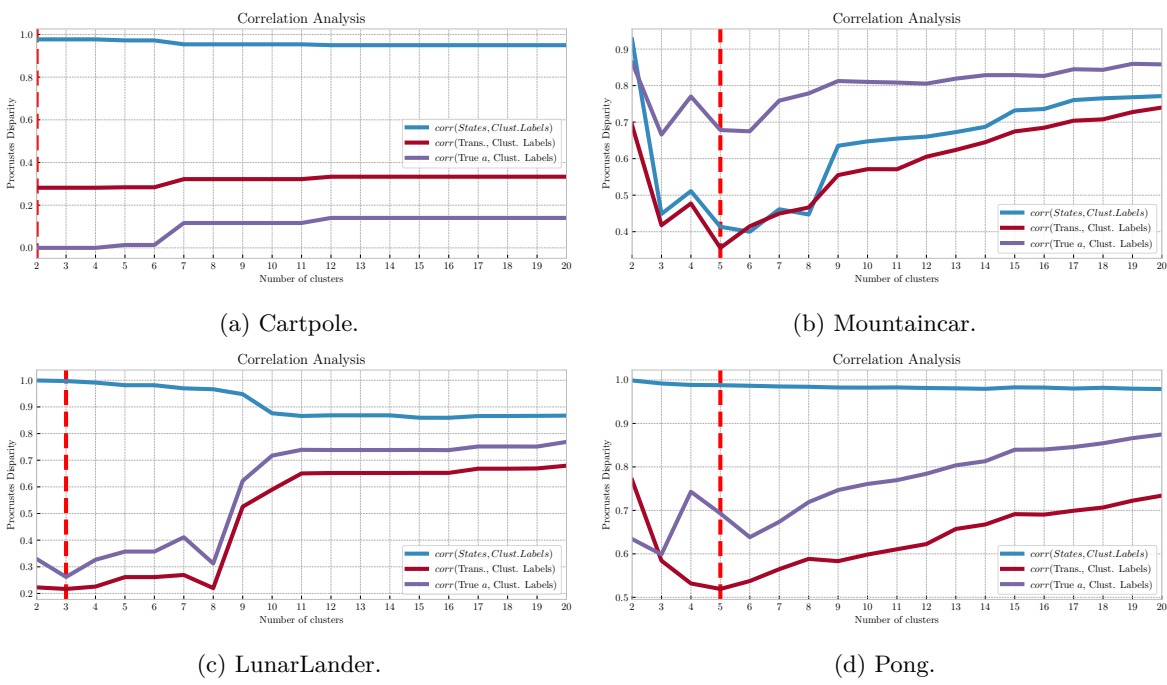

(a) Cartpole.

(b) Mountaincar.

(c) LunarLander.

(d) Pong.

Figure 7: Correlation between the latent action labels obtained from clustering. In the LunarLander experiment the data was from the stochastic environment with deterministic expert. In all other environments the environment was deterministic.

and we can see that baseline ILPO already works well in such environments despite the discontinuities. More complex systems or real-world applications often introduce additional damping or stabilization mechanisms that prevent the occurrence of large-scale dynamic responses. Furthermore, the tools presented in our work can be used to analyze expert data and evaluate the applicability of our pre-training-based approach beforehand.

### 6.3 Selecting the Number of Latent Actions

A critical parameter in ILPO and for our pre-training extension is the number of clusters/latent actions. Edwards et al. (2019) have investigated the effect of the number of latent actions by repeating the experiments and plotting performance against the number of used latent actions. The results show that not all numbers of latent actions lead to good results, and the number of real actions available is used as an initial guess. In the IfO setting, we assume that we have no access to the expert's action data Therefore we can not use this information when choosing the number of latent actions. Instead, we propose a method to identify the number of latent actions/clusters based on how well a clustering given the number of latent actions conforms to the assumption we make on the transition-action similarity.

In Fig. 7, we plot the Procrustes disparity like in table 1 for a range from 2 to 20 latent modes[4]. According to the proposed metric, the optimal number of latent actions/clusters maximizes the similarity (minimizes Procrustes disparity) between transitions and clusters shown in dark red. Those values are shown by a vertical dashed red line in the figures. The quality of this measure can be verified by comparing it to the similarity between cluster labels and ground truth expert actions shown in purple. We can also see that the similarity between expert states and clusters (in light blue) does not work well as an indicator in 3 out of 4 cases.

Furthermore, this method of visualizing the quality of the obtained clusters can be used to predict if a given expert observation-only dataset can be clustered successfully. If the Procrustes disparity between transitions and cluster labels is close to one for any number of clusters, we can assume that clustering was not successful and can not yield a good prior for the pre-training. This holds for example for the Acrobot environment.

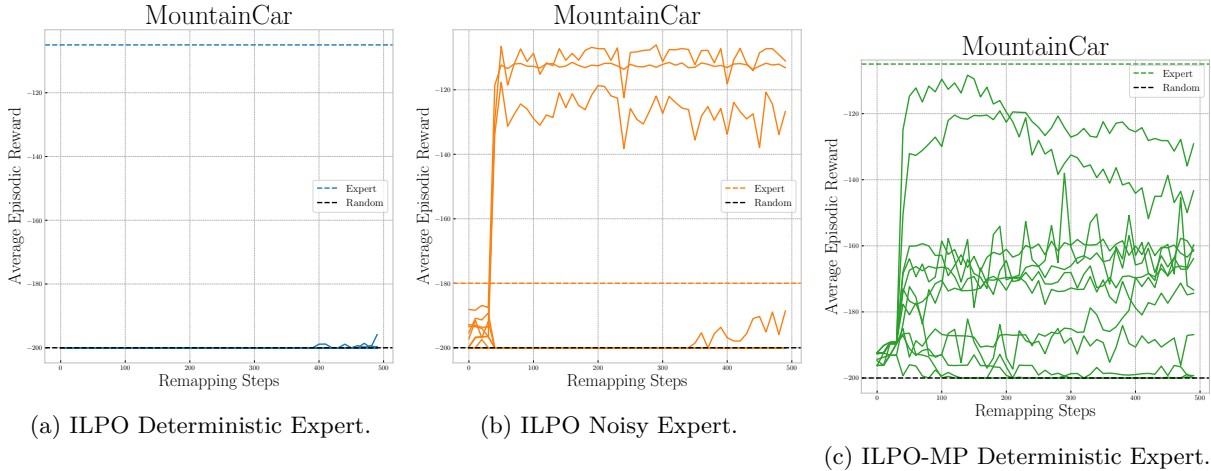

(a) ILPO Deterministic Expert.  (b) ILPO Noisy Expert.

(c) ILPO-MP Deterministic Expert.

Figure 8: Task performance in the MountainCar experiment, plotted for each seed.

### 6.4 Effects of initialisation

As mentioned earlier, baseline ILPO is susceptible to the random initialization of the neural network parameters. The initialization can have a negative impact on the mode separation using the minimum operation in $\mathcal{L}_{\min}$ in Eq. 5 and consequently lead to collapse to one latent action.

$$\forall s \in S : \underset{z}{\arg\min} ||\Delta_t - G_\theta(s, z)||^2 = z^\star \tag{11}$$

One batch $S$ of data points s is passed through the neural network, and $G_\theta(s, z)$ is computed for all possible latent actions z. If, however, all samples in $S$ minimize the loss for the same latent action $z^\star$, we only

---

[4]We omit the Acrobot environment in this analysis as our clustering approach fails in this case, see 6.2

backpropagate through one mode, which is then also much more likely to minimize the loss for the next batch. This effect can accumulate and make mode separation impossible.

As a result, we observe a large spread in task performance for different random initializations. In Fig. 8, we demonstrate this effect in the MountainCar experiment. In Fig. 8a, we can see how ILPO with deterministic demonstrations manages only for one seed by chance to achieve a small reward. With noisy expert demonstrations (Fig. 8b), ILPO manages to solve the task a few times, but most of the time fails to learn the task. With pre-training (ILPO-MP) (Fig. 8c) these catastrophic failures are prevented.

### 6.5 Stochastic Environments

Lastly, we will briefly discuss the issue of stochasticity in the environment instead of the expert demonstrations. We found that repeating the experiments in stochastic versions of the environments will not have much effect on the mode collapse problem. Stochastic environments add noise and make the states more diverse but do not introduce variance in the local action selection that enables mode separation. Fig. 9 shows the MountainCar expert state-action distribution with a deterministic expert policy in a stochastic version of the environment. Compared to the baseline in Fig. 2a, there is more noise around the origin, but the original problem persists.

The performance and latent action diversity of the deterministic policy in the stochastic version of the MountainCar environment exhibits the same collapsed state.

## 7 Conclusion and Further Work

In this work, we investigate the mode collapse problem when imitating latent policies from observations only. We show that mode collapse is caused by a combination of bad initialization and unfavorable properties of the expert demonstrations. To address both issues, we proposed ILPO-MP, a clustering-based method to pre-train latent policy and latent dynamics model. We show that with this pre-training step, we can learn latent policies in environments where it was previously not possible and improve imitation learning performance. Furthermore, we propose a method to automatically decide an important hyperparameter of latent policy learning methods, the number of latent actions.

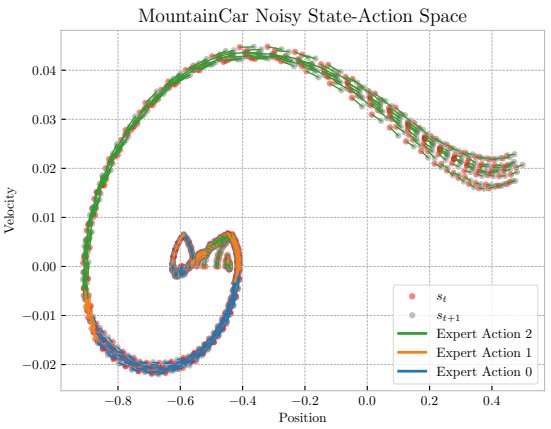

Figure 9: Visualisation of the expert states, actions and transitions in the stochastic MountainCar experiment

In the future, we see potential in further exploring design choices in both latent policy learning and action remapping. In the action remapping step, the exploration mechanism is a key component when obtaining high-quality data from the environments and greatly contributes to the data-efficiency of the method. In the latent policy learning step, more advanced representation learning methods leveraging the temporal smoothness of the expert data or using sequences instead of single states as input could improve the quality of the learned latent policy and dynamics model.

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

# A    Appendix

## A.1    Experiment Details

Our implementation is based on the ILPO implementation by Edwards et al. (2019) and ported from TensorFlow to PyTorch. We aim to publish our code on GitHub after publication.

We used the same neural network architectures and same hyperparameters for latent policy learning and remapping as Edwards et al. (2019). The only change being that we use 50 instead of 100 steps for the latent policy learning. For the pre-training we use 20 epochs for the Acrobot, Cartpole, MountainCar and LunarLander experiment and 30 for the Pong and Pixel Pong experiment.

We used 12 seeds for the latent policy learning and recorded 25 remapping experiments for each latent policy. Every task performance plot is therefore comprised of 300 experiment runs per method. We made sure to maintain the same initialisation of network weights (Xavier uniform for weights and zeros for biases) since we found that using different initialisation, such as the standard PyTorch weight initialisation leads to differences in performance when compared to the TensorFlow implementation. This further strengthens our finding that ILPO is very sensitive to model initialisation.

The Pong environment is a custom environment we developed conforming to the OpenAI Gym API. The environment is available on github. The LunarLander environment in its baseline version is stochastic and the other environments are deterministic. For the experiments, we made stochastic versions of each environment, most notably the MountainCar environment which we discuss in this paper. The stochastic MountainCar environment applies uniform noise to the force applied to the cart.

The number of latent actions used for ILPO was selected based on the number of available actions in the environment. For ILPO-MP we determined the number of latent actions as described in 6.3. Table 2 shows the number of latent actions used for ILPO-MP.

Table 2: Number of latent actions used in the experiments for ILPO-MP.

| Environment | Expert | Latent actions |
|---|---|---|
| Acrobot | Det. Expert | 19 |
| Cartpole | Det. Expert | 2 |
| MountainCar | Det. Expert | 5 |
| MountainCar | Noisy Expert | 3 |
| LunarLander | Det. Expert | 12 |
| Pong (vector) | Det Expert | 6 |
| Pong (vector) | Noisy Expert | 8 |
| Pong (pixel) | Det Expert | 6 |
| Pong (pixel) | Noisy Expert | 8 |

The experts we used have been trained using the PPO implementation (Schulman et al., 2017) from stable baselines3 (Raffin et al., 2021). Deterministic expert data has been recorded by taking the mode of the PPO action distribution and noisy data by uniformly sampling from the action space 20% of the time. Like Edwards et al. (2019) we recorded 50000 samples consisting of state and next state from each expert.

