# OpenReview forum: "ILPO-MP: Mode Priors Prevent Mode Collapse when Imitating Latent Policies from Observations"
_TMLR — Accepted by TMLR_

### Review · Reviewer_QVpZ · 2023-06-23

**Summary Of Contributions:**

- This paper is concerned with tackling the problem of imitation learning from observations where the expert actions are not available from the dataset. This is an important problem because it would enable us to use a vast amount of action-free videos, like YouTube videos, for behavior learning.
- The main contribution of this paper is to improve the previous method for this setup, i.e., Imitating Latent Policies from Observation (ILPO), by (a) identifying the problem of ILPO where latent policies collapse and thus fail to learn useful latents from expert demonstrations and (b) addressing the problem by imposing a mode prior based on clustering of the expert demonstrations.
- This paper provides interesting analysis on how the mode prior can happen per the characteristics of the datasets
- This paper provides experimental results that show the proposed method can improve the performance of ILPO when the datasets are very clean, e.g., collected with deterministic expert policies.

**Audience:**

Yes

**Claims And Evidence:**

No

**Requested Changes:**


Improvement in writing is crucial for recommending the acceptance of this paper. Currently it's not formal and writing makes it difficult to fully understand the paper.
- It's not clear what $\Delta_{t}$ is from the paper. I had to have a look at the original ILPO paper for more formal definition on the notations. From my understanding, G is predicting the difference $\Delta_{t}$ between $s_{t}$ and $s_{t+1}$  instead of directly predicting $s_{t+1}$. Then equation (6) does not make sense, because G that predicts the difference is trained to regress the full state $s_{t+1}$. The paper should add a mention that the notation is a bit misused for clarity like as in original ILPO paper, or should try to make the notation and equation be more formal.
- Understanding this sentence seems very important for the paper but it's really difficult to understand: `Instead, we base our mode prior on the observation that the same actions in the expert data often lead to similar transitions`. What does 'transitions' exactly mean here, and what does the 'same actions in the expert data often lead to similar transitions' mean? How can same $a_{t}$ lead to the similar transition $(s_{t}, a_{t}, s_{t+1})$? After reading all the pages, it seems to me that it meants `same actions often lead to similar state differences`, but it's not clear.
- Above point really makes it difficult to understand the exact meaning and interpretation of the analysis in section 4.2. It's not clear what does 'state' and 'transition' mean here, and what does the distance exactly mean. Adding more explanation on how to interpret the figures in Fig 3 and Fig 4 could be helpful for making the readers understand exactly what's going on.
- Moreover, what does 'Procrustes disparities' mean here? Please provide the intuitive meaning of the distance and also provide the reference, without assuming such knowledge from the readers.
- Explaining the reason behind `We observed that the latent policy learning can end up in a collapsed state despite the pre-training when the latent policy learning is running for many epochs.` should be more helpful for understanding the intuition behind introducing Guided ILPO-MP
- For self-contained writing, please add the reference to agglomerative clustering. And providing more clearer definition and examples for $z^{l}$ should be helpful.

The paper also needs a more support on the assumption on the datasets.
- The paper claims that, unlike the prior work that claimed humans usually collect noisy data, humans are more likely to collect the very clean dataset without noises. But it seems much more natural to assume that humans are likely to collect the noisy data -- for instance, see RoboMimic [Mandlekar et al., 2021] dataset where average humans usually collect very noisy demonstrations and also proficient humans often fail to collect perfectly clean data. The paper can much be more strengthened by providing some experimental evidences to support this claim.
- The paper assumes that dataset follows piecewise continuous dynamics. But first of all, it's not clear what does this exactly mean, so it would be nice to be more formal on this front. Moreover, it's not clear how this would hold for more complex datasets, for instance, locomotion tasks with quadrupeds or humanoids. And this assumption could severly limit the applicability of the proposed method in pixel-based environments except simple environments like Pong where the visual observation is very clean. The paper can much be more strengthened by providing additional experimental results on complex environments. For instance, the paper could provide experimental results on DMC with bang-bang control [Seyde et al., 2021] to see that the method can work on more complex locomotion tasks and this should be feasible because the action space is discrete with bang bang controller. And also the paper could consider conducting additional experiments on CoinRun as in the original ILPO paper to show that the proposed method can work on more visually complex environments.

[Mandlekar et al., 2021] Mandlekar, Ajay, et al. "What matters in learning from offline human demonstrations for robot manipulation." arXiv preprint arXiv:2108.03298 (2021).
[Seyde et al., 2021] Seyde, Tim, et al. "Is bang-bang control all you need? solving continuous control with bernoulli policies." Advances in Neural Information Processing Systems 34 (2021): 27209-27221.

**Strengths And Weaknesses:**

**Strengths**
- The paper tackles an important problem
- Interesting analysis on the effect of dataset for learning latent policies
- Improvement over ILPO on considered tasks

**Weaknesses**
- Paper is not formal and not self-contained as several notations are not formally defined and the paper assumes the knowledge of various techniques without providing the background. This makes it very difficult to fully understand the paper
- Assumption that the clean expert demonstrations is more prevalent is questionable.
- Assumption on the piecewise continuous dynamics is not clearly defined so it's not sure what it exactly means, and this makes it difficult to expect how this proposed method can scale to more complex setups.

**Summary**
The paper tackles an interesting problem and proposes a method based on interesting analysis and it has a substantial potential to be a useful method for this domain. But current writing of the draft makes it fully understand what's going on and what does several components exactly mean. And it's missing empirical support on the important assumptions made in the paper.

---

### Review · Reviewer_BpQB · 2023-06-25

**Summary Of Contributions:**

In this paper, the authors propose a simple approach to alleviating the mode collapse issues when imitating latent policies from observational data only, not relying on expert actions. The main idea is, by clustering expert transitions, to impose a mode prior on the generative model and the latent policy, so that it is less likely to initialise the model leading to model collapse. Experiments on CartPole, LunarLander, and Pong demonstrates the validity of the proposed method.

**Audience:**

Yes

**Broader Impact Concerns:**

The authors do not describe the ethical implications of this work in the paper.

**Claims And Evidence:**

Yes

**Requested Changes:**

See the **Weaknesses** above.

**Strengths And Weaknesses:**

**Strengths:**

+ The paper is very well written and easy to follow.

+ The proposed approach to solving the mode collapse issue when doing imitation learning from observations only is well motivated via a detailed analysis on toy examples, and is also simple to implement.

**Weaknesses:**

- My main concern is how the assumption that "*the same actions in the expert data often lead to similar transitions*" is practical in real world-like applications. Although Table 1 shows the assumption probably holds true in simple RL environments, it is still doubtful if it can be also satisfied in more challenging RL environments with high-dimensional and complicated sensory input. Furthermore, how valid is the assumption especially when the irrelevant backgrounds change in the environments (e.g., section 6.1 of [https://arxiv.org/pdf/2006.10742.pdf](https://arxiv.org/pdf/2006.10742.pdf))?

- At the current stage, the main experiments are only limited to three simple RL environments: CartPole, LunarLander, and Pong. I suggest the authors conduct more experiments and analysis on more complex environments to further demonstrate their assumption and the proposed approach.

---

### Review · Reviewer_bKym · 2023-07-15

**Summary Of Contributions:**

This manuscript focues on understanding and addressing the model collapse issue of a imitation learning from observation algorithm called ILPO. The authors start off with some theoretical analysis on how mode collapse could happen in case of a deterministic expert and bad model initialization, and empirically demonstrate how can the issue be mitigated with the help of a noisy expert. Further, a method called ILPO-MP is proposed to alleviate mode collapse without the need for noisy expert, which could be unrealistic in some tasks. The main idea is to use the cluster labels of the observations (transitions) to anchor the latent policy as a mean of pretraining. Moreover, the authors introduces guided ILPO-MP, which tag the pretraining loss as an auxllilary task during ILPO when the training episodes are long, ex. vision-based learning. There are comprehensive experimental results on several environments to justify the the effectiveness of the proposed method, especially when in absense of noisy demonstrations. Some additional analysis on domains that ILPO-MP does not work (Acrobot), selection of hyperparameters, initialization, environment stochasticity are also provided.

**Audience:**

Yes

**Broader Impact Concerns:**

See [Weaknesses]

**Claims And Evidence:**

Yes

**Requested Changes:**

See [Weaknesses]

**Strengths And Weaknesses:**

[Strengths]

+The paper studies a relevant and significant questions of mode collapse in imitation learning. Although much of the text is about a specific method, ILPO, but I believe it could shed some light of other approaches as well, as action ambiguity has been plagued ILfO, even for IRL-based method like [1], which has also flagged similar issues.

+Detailed analysis of the model collapse problem in the context of ILfO and ILPO, with both empirical evidence and theoretical analysis, motivint the solution pretty well.

+The proposed method, ILPO-MP, which uses data cluster labels to anchor the latent policy, is technically sound and has demonstrated its effectiveness on adressing the mode collapse issue. The method is backed with comprehensive results, and the authors are very transparent on many of their boundaries and limitations (with related results as well). I appreciate it.

+The authors suggest a method to select the optimal number of latent actions/clusters, which is a useful addition to the learning process.

[Weaknesses]

-Although it's ok to focus on a certain approach, how the authors shape the presentation of this paper could potentially impair its impact, as it does not seem clear how can this issue of mode collapse and their solution be elevated to other imitation learning from observation schemes, or imitation learning in general.

-Ovearll, the scope of the experiment section could be a bit limited, as most results are from MountainCar, CartPole, Pong, etc, which are relatively simple for ILfO. This could raise questions on the generalization and real-world performances of the proposed method. The authors are suggested to include more challenging domains, ex. MuJoCo tasks, more Atari games, etc.

-While I agree that a TMLR submission could simply focus on addressing the issue of a certain algorithm, but it would help the readers understand the big picture of the field and the position of the method here if some existing ILfO solutions are compared. Also, since ILPO is posed as a latent variable model learning problem, some other approaches on addressing mode collapse, ex. those from the generative learning community, could be compared here as well.

Minor:

-[1] raises similar issue mentioned in this manuscipt, i.e. ill-posed inverse dynamics and indeed is a concurrent work to Sun et al., 2019. The authors should consider citing this paper.

[1] https://arxiv.org/abs/1910.04417

---

### Decision · Action_Editors · 2023-09-05

**Recommendation:** Accept as is

**Comment:**

The reviewers acknowledge the importance of the problem studied in this paper and the value of the analysis performed by the authors and of the solution they designed to address the mode collapse problem. Nevertheless, the reviewers expressed concerns regarding the clarity of some aspects of the paper, including its scope as well as some more technical notation, and regarding the experimental evaluation, limited to relatively simple RL environments. The authors addressed these concerns in their responses, convincing two reviewers to recommend acceptance (Leaning Accept). The third reviewer failed to provide their final recommendation. The reviewers' main reason to remain on the fence is the experimental evaluation.

The AE believes that the authors have convincingly justified their current choice of evaluation environments, and that there is sufficient value in this work for it to be accepted. As the authors have already updated their manuscript based on their responses to the reviewers, the paper can be accepted as is.

**Audience:**

Yes, the findings of this paper could be of interest to the RL community.

**Claims And Evidence:**

The claims made in the paper are supported by sufficiently convincing evidence. Although the reviewers would have liked to see experiments in more challenging RL environments, they authors explained why such experiments could currently not be conducted and have updated their submission to clarify this accordingly.